# Application of Blockchain Hierarchical Model in the Realm of Rural Green Credit Investigation

**Haoyang Tan \* and Qiang Zhang**

School of Finance and Statistics, Hunan University, Changsha 410000, China; qiangz@hnu.edu.cn
\* Correspondence: tlltwrok@163.com

**Abstract:** In order to realize the application research of blockchain technology in the field of green credit investigation, the current paper adopts the method of a blockchain hierarchical model to study the rural green credit. With regard to the realm of rural green credit investigation, this paper sorts out the characteristics of credit data in China's countryside by countryside credit investigation and determines the major problems and in rural green credit investigation of financial inclusion. Subsequently, the authors put forward a blockchain hierarchical model, which not only has reinforced the advantages in original blockchain dedicated to agriculture, rural areas and rural residents, such as traceability and immutability, but also has transformed the decentralization into disintermediation and changed the single-layered P2P network into a multilayered structure based on China's rural financial environment. Finally, the authors collect and extract the proper credit investigation data on the rural internet to assess the application value of the model by investigating its practical applicability in reality and problems that may occur during the application of the model. Results show that private credit information has an important impact on the prediction accuracy, and the blockchain hierarchical model is helpful to ensure the reliability and security of rural green credit data.

**Keywords:** blockchain; hierarchical model; rural green credit investigation; probability of default





## 1. Introduction

Green credit means that under the guidance of the international "Equator Principle", financial institutions take the environmental responsibility of enterprises as the main basis for providing loans, which guides funds flowing to the green economy and circular economy and promotes the sustainable development of society [1]. The progress of green credit will play a key role in the context of China's economic transformation from high-speed growth to high-quality sustainable development, and the transformation of green development is considered as a core development strategy. Notably, rural green credit investigation has become a tough issue in the credit investigation industry. In the development of rural financial inclusion, we often encounter difficulties such as lack or loss of credit data, incomplete credit investigation infrastructure and poor credit consciousness among farmers [2]. In the advent of the big data era, the demand for farmers' loans is on the rise [3,4]. Therefore, the rural internet credit investigation system will make a great difference in building a good credit environment, expansion of the financial loan for rural small and medium-sized enterprises and promoting rotation of capital and prompt economic development.

By the construction of consortium blockchain, rural green credit investigation data can be used in shared trading. To solve the critical issue in rural green credit investigation system, we can emphasize the study of investigation sharing by means of blockchain technology [5–12]. By virtue of consortium blockchain, we are able to construct a platform of data sharing, high security and openness, on which the trading data in the realm of agriculture, rural areas and rural residents are collected to work out the "data island" problem and enlarge the credit investigation targets so as to lower risk and increase revenue. Current research aims at realizing the characteristics of credit data in China's countryside

by green credit investigation, and figuring out the major problems and, in rural areas, green credit investigation of financial inclusion. Using a sample of 40,000 farmers, authors applied the blockchain hierarchical model to examine credit data evaluation and analysis. Results reveal that private credit information has an important impact on prediction accuracy.

### 1.1. Development, Status Quo and Problems of China's Rural Green Credit Investigation

As of now, the Central Bank credit investigation database has entailed the identity investigation of more than 1 billion people, but the rural green credit records have accounted for less than 20%. In addition, the "data island" problem has permeated in rural green credit investigation data [13], the standard of which is hard to unify. Moreover, the monitory mechanism for credit investigation is incomplete. As a result, it is difficult to effectively integrate data, and thus the use of data is inefficient [14].

The status quo of the credit investigation for rural small and medium-sized enterprises is that most of these enterprises have a problem of expensive and difficult financing. Shorter repayment terms and higher loan interest rates also confine the development of farmers and rural small and medium-sized enterprises. Xu and colleges studied the asymmetric impact of green credit policies and development on the debt financing cost and maturity of different types of enterprises, to quantify the panel data of 52 green enterprises and 81 high-pollution and high-emissions (referred to as "two-high") enterprises in China from 2001 to 2017. With respect to the debt financing cost and maturity, enterprises in economically developed regions are more strongly affected by green credit than those in economically underdeveloped regions [15].

### 1.2. Thinking Analysis of Applying Blockchain Technology on Rural Green Credit Investigation

"Sharing credit investigation" is key to the reform of rural green credit investigation system. With the advent of the big data era, rural green credit data collection has become an approach for rural internet credit investigation, thereby triggering varying problems: difficult and unreliable collection of farmers' credit trading data, insecure information, lack of external exchange, the pervasive "data island" phenomenon and difficulty in supervision. In the current rural green credit investigation industry, how to share farmers' credit information has become the core issue for the credit investigation system and even for the credit system development. China's rural green credit investigation system should use the mature foreign system for reference. The idea of "sharing credit investigation" is a guideline for our system because it caters to China's construction of credit investigation model under the policy for agriculture, rural areas and rural residents and proffers a direction for China's credit investigation system. The impact of information exchange on the credit market depends on the structure of the credit market. The higher the degree of market information monopoly is, the more efficient the mechanism of information sharing becomes [16–18]. In the hierarchical model set up by Kevin and Lindsay Colvin [19], it is presumed that the financial institution is the information monopolist. Under the circumstance of asymmetric information, a failure of sharing credit investigation on the first layer will generate continual competition among institutions on the second layer [20], thereby reducing the cost for borrowers and prompting them to repay the loan.

According to the research of Belotti and other scholars, a more sufficient information sharing will bring about a better precision to the credit risk management and thus uplift the loan quality; besides, it benefits the vulnerable groups, realizes equal credit opportunity and propels rural financial inclusion [21–23]. Therefore, in the context of Big Data Internet, we set forth the "sharing credit investigation" as a solution for problems such as the asymmetric information in financing credit and cost increase caused by varying data sources and difficulty in data collection. Liu proposed a fusion of double blockchain reference data storage and query scheme, which is composed of two chains: one chain is used to store the real-time reference data of many, and the other chain is used to store personal credit report. Double blockchain integration scheme could quickly and automatically generate a personal credit report, avoiding queries privacy risk, tampering and forgery of personal

credit reporting data in the process of credit reporting [24]. In addition, Sahoo and colleges achieved successful application of hierarchical and abstraction-based blockchain model in 2019, which provides a tunable precision in various abstract domains and guarantees the soundness of the system.

This paper provides a reference for the application of blockchain technology in rural green credit investigation through the blockchain hierarchical model. Rural credit investigation in China currently faces a series of problems such as "data island", "data security" and "regulatory difficulty". "Sharing credit investigation" can give full play to the advantages of credit investigation. We attempt to set up a "needless trust" credit investigation mechanism. In such a context, we notice that the nature of blockchain is a new database solution that is decentralized, de-trust, open and autonomous. In theory, these operational characteristics of blockchain all contribute to the application of a "needless trust" credit investigation system. Therefore, this paper combines blockchain with credit investigation, relies on the advantages of blockchain technology to build a blockchain hierarchical model that conforms to China's environmental policies and combines with the real demands of rural green credit investigation. Combined with the empirical study, the feasibility of applying the blockchain hierarchical model to the rural green credit investigation is discussed. In particular, it is concluded that the blockchain hierarchical model reveals effectiveness in predicting the accuracy of default behavior, and the prediction accuracy would vary according to privacy indicators. The present research provides valuable theoretical foundation for the sustainable development of green finance in China.

## 2. Materials and Methods

### 2.1. Rural Green Credit Investigation Hierarchical Model Structure Design

As its name implies, hierarchical blockchain is made up of and run by fundamental chain and subchains simultaneously. To begin with, we divide rural internet credit investigation data into four categories as per sources and sectors: data from licensed financial institutions, data from internet financial technology companies, data from governmental sectors and data from telecommunication and transportation service sectors; next, we sub-divide the four categories based on their data type and realm of business.

Meanwhile, in terms of privacy hierarchy, we can also divide rural green credit investigation data into four categories. The first is strict individual privacy. The second category of data is called structured personal characteristic identifier, which is not exclusive but is very personal. The third category is data shared or associated by an individual with other parties, which has personal characteristics but has no competence in identification. The fourth category is group or overall data and derived data of other parties.

### 2.2. Data Characteristics

Next, by taking into account the fairness of rural green credit investigation data sharing and the validity of incentive and supervision, we believe that comparable credit investigation institutions will probably form a more efficient consortium. Therefore, the structure of blockchain hierarchical model can be set up with three levels [25–27]. The first level includes four categories of data sources (cloud level); the second level entails consortium data from large institutions of internet, financial technology and credit investigation demand (fog level); the third level contains consortium data from small institutions with credit investigation demand (edge level). A small consortium can join the large consortium as an independent node so as to secure the valid sharing and use of credit information data on the internet [28]. The fundamental chain (based in the governmental credit information center), i.e., cloud level in the model, is sub-divided into four major subchains as per four categories of the data source to compile and analyze all the credit information on the internet. This is shown in Figure 1:

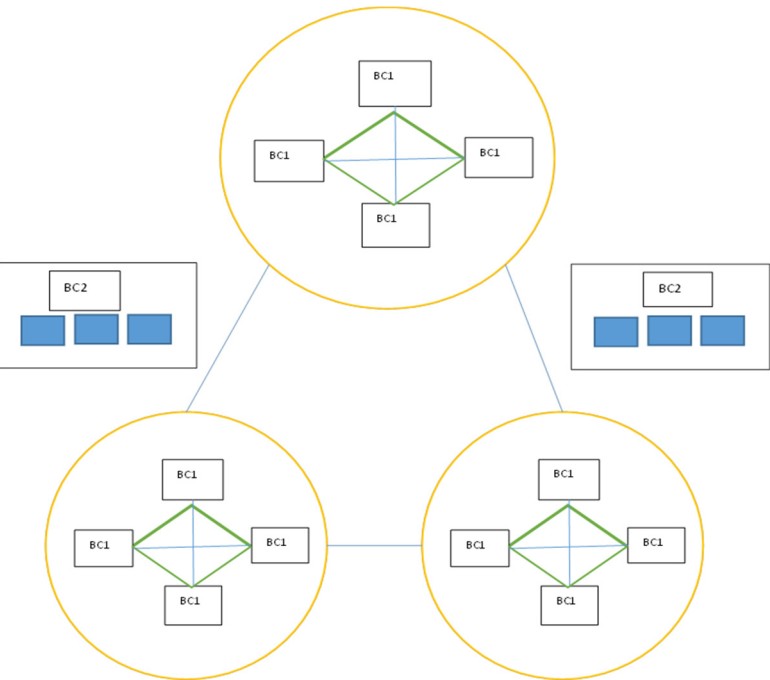

**Figure 1.** Structural map of hierarchical blockchain nodes.

When a credit event happens, data will naturally be collected and packaged by corresponding institutions on the internet platform based on different scenarios where data is generated. Next, under one circumstance, with the help of a smart contract, packaged data blocks are compiled into the original small consortium chain by a certain consensus mechanism [29]. Then, also confined by a smart contract, the small consortium chain becomes a branch of a congenial large consortium chain through consensus; under another circumstance, packaged data blocks are directly compiled into a congenial large consortium chain through certain consensus and with the help of corresponding smart contract. Finally, as per the division of data types on the internet [30], the large consortium chain, with the stimulation of smart contract and proper consensus, will link to the corresponding subchain of the fundamental chain (cloud level). Finally, data will be analyzed, reviewed and monitored on the fundamental chain (cloud level) and the integrity of which will determine the green credit rating of an individual.

## 3. Results

### 3.1. Empirical Analysis

Assume that the interval domain is defined as $\amalg$ = {[l, h] | l∈Z ∪{−∞}, h∈Z ∪ {+∞}, l ≤ h}∪⊥, given a set of integer X, which can be reduced to interval [l, h] (1 is the minimum and h is the maximum). For example, integer group [2, 1, 100, 4] can be reduced to [1, 100].

Suppose $L_C \leq \xi(R)$, is the concrete domain of the numerical R's power, $L_{\amalg} \leq \amalg$, is the corresponding abstract domain of Y; then

$$[l_1, h_1] \subseteq [l_2, h_2] \Leftrightarrow l_2 \leq l_1 \wedge h_2 \geq h_1 \tag{1}$$

$$[l_1, h_1] \cap [l_2, h_2] = [\max(l_1, l_2), \min(h_1, h_2)] \tag{2}$$

$$[l_1, h_1] \cup [l_2, h_2] = [\min(l_1, l_2), \max(h_1, h_2)] \tag{3}$$

The correlation between $L_C$ and $L_{\amalg}$ is defined by Galois Connection $<L_C, \alpha_{\amalg}, \gamma_{\amalg}, L_{\amalg}>$, in which $\forall S \in \xi(Z)$ and $\forall \overline{v} \in \overline{\amalg}$:

$$\alpha_{\amalg}(S) = \perp \quad \text{If } S = \phi \tag{4}$$

$$\alpha_{\amalg}(S) = [l, h] \quad \text{If round down}(\min(S)) = I \wedge \text{round up}(\max(S)) = h \tag{5}$$

$$\alpha_{\text{II}}(S) = [-\infty, +\infty] \quad \text{If nonexistent } \min(S) \wedge \text{nonexistent } \max(S) \tag{6}$$

$$\alpha_{\text{II}}(S) = R \quad \text{if } \overline{V} = [-\infty, +\infty] \tag{7}$$

$$\alpha_{\text{II}}(S) = \{k \in R | l \le k\} \quad \text{if } \overline{V} = [l, +\infty] \tag{8}$$

**Blockchain in Abstract Domain of Character**

In this chapter, we will provide an abstract domain for character string data. Although there are many kinds of abstract domains, we review two of them: "prefix and suffix" and "block". Given a letter $\Sigma$ made up of characters and a character string is the sequence of elements in $\Sigma$ (which may be infinite), let us suppose $S = \Sigma^*$, and it is a collection of all possible strings of any length (which also may be infinite) generated by $\Sigma$. A concrete character string domain is presented in the form of grid: $L_C \, L \, \xi(S), \subseteq, \varnothing, S, \cup, \cap >$.

**Abstract Domain of Prefix and Suffix**

Prefix x is the abstraction of a group of character strings (including character string x itself), with x as the beginning, followed by other characters. In form, abstract domain of the prefix is demonstrated by the abstract domain $L_A \le A$, in which $A = \Sigma^* \cup \{\bot\}$. Given two prefixes $\overline{x}, \overline{y} \in A$, partial order $\subseteq$ is defined as:

$$\overline{x} \subseteq \overline{y} \text{ if } \text{len}(\overline{x}) \ge \text{len}(\overline{y}) \wedge \forall i \in [0, \text{len}(\overline{y}) - 1] : \overline{x}[i] = \overline{y}[i] \tag{9}$$

In which len (a) returns to the number of characters in "a". In other wordss, if y is the prefix of x, y will be greater than x. We see that the top element in A is an empty prefix, so operation $\cap$ and attended operation $\cup$ are defined as:

$$\cap \, (\overline{x}_1, \overline{x}_2) = \overline{x}_1 \quad \text{if } \overline{x_1} \subseteq \overline{x_2} \tag{10}$$

$$\cap \, (\overline{x}_1, \overline{x}_2) = \overline{x}_2 \quad \text{if } \overline{x_2} \subseteq \overline{x_1} \tag{11}$$

$$\cup \, (\overline{x}_1, \overline{x}_2) = \overline{x}_1 \quad \text{if } \overline{x_2} \subseteq \overline{x_1} \tag{12}$$

$$\cup \, (\overline{x}_1, \overline{x}_2) = \overline{x}_2 \quad \text{if } \overline{x_1} \subseteq \overline{x_2} \tag{13}$$

Therefore, The Galois Connection between $L_C$ and $L_A$ is established as $<L_C, \alpha A, \gamma A, LA>$, in which:

$$\forall X \in \xi(S) : \alpha_A(X) = \cup_{x \in X} \alpha(x) \text{ or } \alpha(x) = x \tag{14}$$

$$\forall \overline{v} \in A : \gamma_A(\overline{v}) = \{s \in S | \text{prefix}(s) = \overline{v}\} \tag{15}$$

*3.2. Calculation Process*

For experimental purposes, we have collected loan events of more than 40,000 farmers (with a scope of between 100 and 2000). In the experiment, we randomly selected a testing dataset of 500 farmers' credit trading records and divided it into two parts: existent data in a concrete blockchain and nonexistent data in a concrete blockchain. We then searched the dataset by changing latency time and transaction rate [27] and gained the results in Table 1.

Although PDR is high in the current results, we can lower it by adjusting abstract model and reducing latency time $\delta$. By virtue of an abstract explanatory framework, we can use abstract models of all levels, from coarse grit to fine grit. For example, as per the abstract case in the interval domain, integer x can be abstracted as [x,x] or [$\infty$,$\infty$]. In addition, if the framework is robust enough, we can take all kinds of abstract domains (domain of relation, domain of non-relation or their combination) into account. In general, the lowest-level blockchain represents farmers' loan data, whereas a higher-level blockchain stands for a higher abstract value and abstract degree. Therefore, as we march towards a higher level, though the empirical results appear to have increased, their validity actually falls accordingly. The framework allows people to execute verification at the proper hierarchical level, which relies on the accuracy required in the empirical results, or to execute verification in a specific blockchain of the lowest level through the entire hierarchical structure.

**Table 1.** Comparison of Empirical Results of EDR and PDR.

| Target Layer | δ | Rate | Verification Results in Abstract Blockchain | | | | |
| --- | --- | --- | --- | --- | --- | --- | --- |
| | | | No. of Inputs for Verification | No. of Inputs | Exists in CB? | % of EDR | % of PDR |
| Comparison of Empirical Results of EDR and PDR | 2 | 0.8 | 500 | 250 250 | Ture False | 25.2 | 46 |
| | | 1 | 500 | 250 250 | Ture False | 20.2 | 48.8 |
| | | 1.25 | 500 | 250 250 | Ture False | 13.4 | 49.6 |
| | 5 | 0.8 | 500 | 250 250 | Ture False | 19 | 48 |
| | | 1 | 500 | 250 250 | Ture False | 14.2 | 49.4 |
| | 7 | 1.25 | 500 | 250 250 | Ture False | 17.2 | 49.8 |
| | | 1.4 | 500 | 250 250 | Ture False | 18.2 | 49.9 |
| | | 0.8 | 500 | 250 250 | Ture False | 12.8 | 49.2 |
| | 9 | 1 | 500 | 250 250 | Ture False | 9.2 | 50 |

In this section, we analyze the repercussions of the prohibition against using private information on the efficiency of internet credit resource allocation, so that there are only two values of the dependent variable, namely "breach of contract" and "abide by contract", and thus we adopt a Logistic Regression Model. The reason is that the explanatory variable in Logistic Regression Model is a dummy variable, meaning the variable only includes 0 and 1 to indicate the occurrence or nonoccurrence of an event, in correspondence to "breach of contract" and "abide by contract" herein; besides, the predictive variables in Logistic Regression Model can be continuous or dummy, and they are not required to be in normal distribution; plus, the Logistic Regression Model has been applied in a number of credit risk analysis cases of both home and abroad, which proves the feasibility of this regression method.

By means of Python data capture software, we have extracted 20,000 pieces of rural credit customer information on agricultural information websites and obtained 12,000 complete and valid data after deleting deficient data with incomplete fields. These data represent 12,000 rural credit customers, including 8000 "good customers" and 4000 "bad customers". This entails 14 variables associated with the borrower, such as the farmer's current account status, deposit account status, account period, loan history, loan amount, loan usage, farmer status and contact method. The 14 variables include 3 continuous variables, 4 discrete variables and 7 character variables, which basically cover the individual's basic information, economic status, credit history and credit status quo.

Considering the data are very complete, we omit the step of removing clutter and noise. However, in order to enhance the model efficiency, we still normalize the numeric variables in the original data. We also quantize character data so as to apply sorting algorithms.

The experiment is divided into two tests. The first is to test the accuracy of default prediction with complete internet information data samples; the second is to test the accuracy of default prediction after individual status, gender, deposit account status and financial status are removed from the data samples. In contrast with the two test results, we can analyze whether the prohibition of partial private information will lower the accuracy of a personal credit model and thus influence the rural internet credit investigation.

Computational results of data with complete internet credit information are shown in Table 2.

**Table 2.** Accuracy of Farmer's Default Prediction with Complete Information.

| Test Result | Number | Percentage |
|---|---|---|
| Correct Prediction | 9317 | 77.6% |
| Wrong Prediction | 2683 | 22.4% |
| Total | | 100% |

Computational results of internet credit data when partial information is prohibited are shown in Table 3.

**Table 3.** Accuracy of Farmer's Default Prediction with Partial Information.

| Test Result | Number | Percentage |
|---|---|---|
| Correct Prediction | 9056 | 75.5% |
| Wrong Prediction | 2944 | 24.5% |
| Total | | 100% |

From Tables 2 and 3, we can see that when the internet information is complete, the accuracy of default prediction in the credit evaluation model is 77.6%, whereas when private information "individual status and gender", "deposit account status" and "financial status" is prohibited, the accuracy of default prediction drops by 2.1%.

Table 4 illustrates the significance of each characteristic variable on the prediction of dependent variable:

**Table 4.** Significance of Each Characteristic Variable on Default Prediction.

| Index System | Group | | | | |
|---|---|---|---|---|---|
| | 1 | 2 | 3 | 4 | 5 |
| Current account status | 0.35 | 0.35 | 0.3 | 0.29 | 0.3 |
| Loan history | 0.25 | 0.25 | 0.2 | 0.18 | 0.2 |
| Financial status | 0.1 | 0.09 | 0.1 | 0.09 | 0.08 |
| Individual status and gender | 0.05 | 0.05 | 0.05 | 0.05 | 0.05 |
| Percentage of installment in monthly income | 0.03 | 0.03 | 0.03 | 0.03 | 0.03 |

From Table 4, we can see "current account status" and "loan history" play a great role in default prediction, whereas credit information such as "financial status", "individual status and gender" and "percentage of installment in monthly income" also make a difference in default prediction, though their significance is less than the former two kinds of credit information.

According to the 14 variables collected, the results show that household credit information has an important impact on the prediction accuracy, and the blockchain hierarchical model is helpful to ensure the reliability and security of rural green credit data.

## 4. Discussion

Research conducted by Liu shows that Chinese financial institutions are still in the primary stage of integrating environmental and social factors into their loan policies, and the development of green credit in Africa is faced with the problem of improving the ability to manage overseas risks and environment [31]. The same problem is shared by authors who aim to improve the reliability of rural credit data evaluation. It is expected that the blockchain hierarchical model would be used to further screen more green credit data. The empirical results show that some private personal credit information has great significance on the personal credit rating on the internet, which is consistent with previous research on credit rating models [32]. When such information is prohibited, the accuracy of default prediction in an internet credit rating model will decrease and influence the

credit risk evaluation of credit applicant as well as the efficiency of credit resource allocation. If we apply gender and other privacy variables into the credit evaluation model, more borrowers with good credit will be granted with loans, and the individual credit investigation market will be enhanced accordingly. Therefore, information security and credit information sharing environments based on a blockchain hierarchical model can be achieved and will benefit the development of an internet credit market theoretically. This model will guarantee the reliability, security and traceability of data in rural green credit investigation, which complies with agricultural information construction. It also demonstrates that the blockchain hierarchical model is of application value in rural internet credit investigation market.

## 5. Conclusions

There is no doubt that the development of rural green credit is hardly separated from the joint efforts of government regulatory authorities and commercial banks and the support of corresponding financial technology. The present research focuses on investigating data characteristics of China's rural green credit, capitalizing on blockchain hierarchical model to realize data evaluation and analysis, which highlights the prominent effectiveness of the FinTech approach when analyzing massive and complex data regarding green credit. Presently, the implementation plans of green credit policies in a number of major countries, such as the United States, the United Kingdom and Japan, are jointly formulated by their environmental protection departments, relevant commercial banks, key enterprises in the industry, industry associations and industry experts. Currently, China has piloted green credit information databases in some regions, and the hierarchical management, dynamic exchange and resource sharing of rural green credit data information could be accurately realized by a blockchain hierarchical model [33,34].

The current study provides empirical evidence that private credit information has an important impact on prediction accuracy, and the blockchain hierarchical model is helpful to ensure the reliability and security of rural green credit data. The study also reveals that "current account status" and "loan history" show significant influence in default prediction. The present research is a meaningful exploration of the application of the blockchain hierarchical model in rural green credit data analysis, which could be used as a basis for further study with respect to the impact of blockchain technology on environmentally sustainable development. For further research, the authors suggest focusing on the additional features of the blockchain hierarchical model and how they could affect the accuracy and efficiency of financial institutions in rural environmental assessment, environmental risk tracking after green credit supply, and green audit. Futhermore, the generalizability of the research is expected to be achieved by extracting and analyzing information in different countries.

**Author Contributions:** Conceptualization, H.T.; methodology, H.T.; software, H.T.; validation, H.T.; formal analysis, H.T.; investigation, H.T.; resources, H.T.; data curation, H.T.; writing—original draft preparation, H.T.; writing—review and editing, H.T. and Q.Z.; visualization, H.T.; supervision, H.T.; project administration, Q.Z.; funding acquisition, H.T. and Q.Z. All authors have read and agreed to the published version of the manuscript.

**Funding:** This research was funded by the Science Fund for Creative Research Groups of the National Natural Science Foundation of China, grant number 71661137006, and the Innovation Fund for University Production, Study and Research of the Ministry of Education Science and Technology Development Center of China, grant number 2019J02015.

**Institutional Review Board Statement:** Not applicable.

**Informed Consent Statement:** Not applicable.

**Conflicts of Interest:** The authors declare no conflict of interest in the collection, analyses, or interpretation of data, in the writing of the manuscript, or in the decision to publish the results.

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
