# Peer review of "Application of Blockchain Hierarchical Model in the Realm of Rural Green Credit Investigation"

_sustainability, doi:10.3390/su13031324_

Round 1
Reviewer 1 Report
Dear authors,
Your paper deals with an interesting topic: rural green credit investigation. However, in my opinion you should improve it.
Concerning introduction, please refer the research problem and the main purpose of the article. Moreover, link the introduction witk literature review (you only add references in line 35). Moreover yet, present in the introduction the structure of the paper.
With regard to the literature review, the weight of section 1.1. and 1.2 is not similar. In section 1.1 some references are missing (see for instance line 43 and 48, 49). Can you please clarify and define what is “rural green credit investigation”?
In section 1.2 in my opinion the aformentioned problem is constant. Please give more attention in citations and references (line 52, 70, etc.)
Concerning methodology, can you clarify here what is the model that you use (logit model?) or for instance the software???
Concerning, results please revise this section. Here, I think that the content that you wrote is much more relate with methodology than with results. See for instance the section 3.1.1.
In the Discussion section, revise this section according literature review.
Last but not the least, add a section of conclusions. Here refere what are the implications for theory and practise. Also add the limitations of the study and lines for future research.
Reviewer 2 Report
General
The article under review promises to be very interesting. The article discusses interesting issues related to the Application of Blockchain Hierarchical Model in the 2 Realm of Rural Green Credit Investigation. I would like to highlight some comments that will hopefully help the authors to further improve the work.
Overall, the article needs more order and clarity. Please note the structure of the article: Abstract, Introduction, Literature review (not included in the article), Materials and Methods, Results, Discussion, Conclusions (not included in the article), References
I only draw your attention to the need for careful editorial correction.
Please specify the originality of the presented research. Please indicate how the results of the analysis can be used? How do the results advance the current knowledge? Whether and to what extent the result for other countries can be used.
Please extend the bibliographic references in part of the discussion to possible publications
Abstract
It needs to be put in order. What is the value of the research, how the research is carried out (methodology), the main results and conclusions from the research.
No indication of the purpose of the study.
No overall analysis results.
No reference to practice.
Instroduction
The analyzed problem has been briefly presented. In the introduction, there is a place for the purpose of the work (the reviewer does not notice it) and a description of the legitimacy of taking up a given topic (indicated quite generally). No indication of the years of the study or why it will use the indicated methodology. What are the arguments for using the adopted methodology? and why the indicated research area was adopted.
The lack of research questions or research hypotheses makes it difficult to evaluate both this part of the work and it as a whole.
Review literature (not available)
Literature review and research background require more attention. Its element was found in the introduction. It seems that it should be expanded to include international literature (concerning credits, their assessment, reporting).
Materials and Methods
Information on the hierarchical structure model (which the authors use) should be organized
Please indicate which individual variables the authors used in the study.
The letters used in the formulas should be described in their interpretation
Results
The indicated research results require more detailed discussion in relation to the topic under consideration. Some need to be sorted out.
Element 3.11, according to the reviewer, should be part of the material and metda chapter
It seems that lines 124-158 should not be included in this section
There are no sources for tables
The authors point to the logistic regression model, but in the methodological part it is not described
The authors also point to the normalization of the variables / no reference to its method, the methodology did not specify which normalization model was used
Discussion
The presented discussion seems to be a consequence of the introduction, a literature review. In the discussion, the obtained results should be compared with similar analyzes by other authors.
In the opinion of the reviewer, this part only refers to own research results. Lack of their interpretation for other studies (this also indicates the need to expand the literature review)
Conclusions (not yet available)
None in the peer-reviewed study. You can indicate why this study is unique? What are the drawbacks and uncertainties of this study? What have we learned from this study? Benefits for decision makers? Benefits for stakeholders?
Round 2
Reviewer 1 Report
Dear Authors,
Thank you to improve you paper. However in my opinion you should to improve it a little more.
Please define in the introduction section the research problem and the structure of the paper.
Please revise the conclusion section. In my opinion some text belong to metodology and are not results (see for instance the section 3.1.1).
In the conclusion section present the limititations of the study, implications for theory and practise. Also here, link your comments with literature review and add some references.
Good work.
